# The Phase Structural Evolution and Gas Separation Performances of Cellulose Acetate/Polyimide Composite Membrane from Polymer to Carbon Stage

**DOI:** 10.3390/membranes11080618

**Published:** 2021-08-12

**Authors:** Haojie Li, Shan Xu, Bingyu Zhao, Yuxiu Yu, Yaodong Liu

**Affiliations:** 1Key Laboratory of Carbon Materials, Institute of Coal Chemistry, Chinese Academy of Sciences, 27 Taoyuan South Road, Taiyuan 030001, China; lhj1241874828@163.com (H.L.); zby20210810@163.com (B.Z.); yuyuxiu@sxicc.ac.cn (Y.Y.); 2University of Chinese Academy of Sciences, 19 Yuquan Road, Beijing 100049, China; xushan@sxicc.ac.cn; 3State Key Laboratory of Coal Conversion, Institute of Coal Chemistry, Chinese Academy of Sciences, 27 Taoyuan South Road, Taiyuan 030001, China; 4Center of Materials Science and Optoelectronics Engineering, University of Chinese Academy of Sciences, Beijing 100049, China

**Keywords:** cellulose acetate, polyimide, chase separation behavior, carbon molecular sieve membranes

## Abstract

Blending and heat-treatment play significant roles in adjusting gas separation performances of membranes, especially for incorporating thermally labile polymers into carbon molecular sieve membranes (CMSMs). In this work, cellulose acetate (CA) is introduced into polyimide (PI) as a sacrificial phase to adjust the structure and gas separation performance from polymer to carbon. A novel result is observed that the gas permeability is reduced, even when the immiscible CA phase decomposes and forms pores after heat treatment at 350 °C. After carbonization at 600 °C, the miscible CA has changed without contribution, while the role of the immiscible CA phase has changed from original hindrance to facilitation, the composite-based CMSM at a CA content of 10 wt.% shows highest performances, a H_2_ permeability of ~5300 Barrer (56% enhancement) with a similar H_2_/N_2_ permselectivity of 42. The structural analyses reveal that the chain interactions and phase separation behaviors between CA and PI play critical roles on membrane structures and gas diffusion, and the corresponding phase structural evolutions during heat treatment and carbonization determine gas separation properties.

## 1. Introduction

Membrane separation technology, which is more energy efficient than traditional gas separation technologies such as pressure swing sorption and cryogenic distillation, has attracted widespread attention and is rapidly developing [1,2]. The development of membrane-based gas separation technology mainly focuses on achieving superior performances and simple preparation. Many types of polymer-based membranes have hitherto been industrialized and widely used in many industries. However, due to the flexible nature of polymer chains, the trade-off boundary between gas permeability and selectivity is hard to be crossed for polymer membranes, which strongly limits the real separation efficiency [3,4].

Polyimide (PI) could be transformed into carbon materials after pyrolysis. As compared with polymer membranes, carbon molecular sieve membranes (CMSMs) exhibit considerable advantages in gas separation performance and could easily break through the trade-off boundary limit [1,5,6,7]. The superior performances of CMSMs are ascribed to their unique rigid dual-channel structures. During thermal treatment, polymer precursor chains decompose and rearrange, ultra-micropores (<0.7 Å) and micropores (7–20 Å) are formed. The micropores act as fast gas transmission channels, and ultra-micropores separate different types of gas molecules [8,9].

The facial preparation technology, high gas separation performances, and low cost are essential for the applications of CMSMs. In order to improve the gas permeability and selectivity of CMSMs, precursor types and pyrolysis conditions (heating rate, temperature, atmosphere, duration) have been intensively studied [10,11,12,13]. Additionally, the sacrificial component has been adopted to introduce macropores or channels in CMSMs, which improves gas flux [14,15,16,17]. Polyvinylpyrrolidone (PVP) and poly(ethylene glycol) (PEG) are widely used as pore-forming agents due to their low thermal decomposition temperature and low residual. Kim et al. prepared PI/PVP-blend-based CMSM and obtained a maximum gas permeability of 630 Barrer for O_2_ and the O_2_/N_2_ permselectivity of 10 [18]. Some improvements were also observed in polyphenylene oxide/PVP-blend-based CMSMs [19,20]. The phase diagram and structures are essential for tuning the porous structures of polymer-blend-based CMSMs. The effects of molecule weight, which directly affects the phase structures of polymer blends, have been explored [21,22,23], and it is found that a higher molecular weight of the sacrificial phase favors the formation of gas diffusion voids or channels with larger sizes [24,25]. Moreover, the porous structures in CMSMs could be also formed with tunable size distributions by changing the contents of components [15,26,27,28]. For precursor membrane, Giel et al. prepared the polyaniline (PANI)/polybenzimidazole blends, and the gas permeability decreases at a higher PANI content [29]. In contrast, the gas permeability of the PEG/PI membrane increases when the content of low molecular weight PEG increases [30]. The phase diagram and chain interactions of polymer blends are important for the gas permeability of the precursor membrane [30,31,32,33] and are expected to affect the gas separation performances after sequential thermal treatment and carbonization. Thus far, little attention has been paid to how the phase diagram and chain entanglements affect the structural evolution and separation performances of the membrane during thermal pyrolysis and carbonization.

In this work, CA was selected as a sacrificial component in PI-based membranes. In addition to phase separation, a small amount of CA was found to be miscible in the PI matrix at a low CA content, which significantly reduced the gas permeability of the PI membrane and heat-treated PI membrane. The benefits of pores only played major roles after carbonization at 600 °C. These works would be useful for better understanding the relationships between the physical structures of a composite membrane and its gas permeability from precursor to carbon membranes and would solve the puzzles that the gas permeability decreases for polymer membranes after mixing with some sacrificial components. For membrane separation, both chemical and physical factors play roles in the gas permeability and selectivity, especially for composite membranes.

## 2. Materials and Methods

### 2.1. Materials

The PI, 6FDA-DAM:DABA (3:2), was provided by the group of Prof. Nanwen Li in the Institute of Coal Chemistry, CAS, China. The corresponding synthesis steps and ^1^H NMR spectrum of PI are included in Appendix A in Appendix A. Cellulose acetate (CA) was purchased from Aladdin Industrial Co., Ltd. (Shanghai, China). N-Methyl-2-pyrrolidone (NMP) and methanol were purchased from Sinopharm Chemical Reagent Co., Ltd. (Shanghai, China).

### 2.2. Preparation of Membranes

A certain amount of PI powder was dissolved in NMP at 60 °C and continuously stirred for at least 24 h to obtain a homogeneous solution. Then, CA powder was slowly added into PI solution at various PI/CA mass ratios (100:0, 99.5:0.5, 99:1, 95:5, 90:10, 80:20). Finally, the composite solution was further stirred at 70 °C for 24 h to ensure complete dissolution, and the total solid content was 20 wt.%. The sample compositions and abbreviations are summarized in Table 1.

For membrane preparation, the solution was cast on a clean flat glass plate using a casting knife with a gap thickness of 200 μm. The cast membranes were immediately dried at 60 °C for 1 day. Then, the dried membranes were peeled off from glass and immersed in room temperature methanol for 24 h. Finally, the membranes were flattened and vacuum-dried at 150 °C for 24 h to completely remove the residual solvent. The thickness of the obtained membranes was ~20 μm.

### 2.3. Heat Treatment

The polymeric membranes were heat-treated in a tubular furnace with a continuous nitrogen purging flow of 200 mL/min. The tube was purged by nitrogen for more than 6 h before heating to remove oxygen. The heating profiles were from room temperature to 280 °C at a ramp rate of 5 °C/min; next, from 280 °C to 300 °C at a ramp rate of 1 °C/min; then, from 300 °C to 350 °C at a ramp rate of 0.5 °C/min, and isothermal for 60 min. For the preparation of heat-treated PI films, the furnace was naturally cooled down to room temperature and the membranes were collected for further characterizations. For the preparation of CMSMs, the temperature was further increased to 585 °C at a ramp rate of 3 °C/min; then, from 585 °C to 600 °C, at a ramp rate of 0.25 °C/min, and isothermal for 120 min. During the heating process, the membranes were pressed by graphite blocks to prevent membranes to become curly.

### 2.4. Characterization

The synthesized PI was analyzed by NMR experiments (model: Bruker Avance III, Karlsruhe, Germany). The prepared membranes were monitored by a Fourier transform infrared spectrophotometer (FTIR, model: Bruker TENSORII, Karlsruhe, Germany) to obtain infrared spectra, 128 scans were token from 4000 cm^−1^ to 700 cm^−1^ at a resolution of 2 cm^−1^. The thermogravimetric behavior was recorded by thermogravimetric analysis (TGA, model: Mettler Toledo TGA-2, Zurich, Switzerland). The morphological structure of surface and cross section and EDXS mapping were observed by a field emission scanning electron microscope (SEM, model: JSM-7900F, Tokyo, Japan) and element analyzer (EDXS, model: Vario EL CUBE, Hanau, Germany). The phase structure of composite membranes was observed by optical microscope (OM, model: OLYMPUS CX31, Tokyo, Japan). The glass transition temperature was measured by a dynamic mechanical analyzer (DMA, model: NETZSCH 242-E, Selb, German). CA and PI before and after treatment were characterized by a Bruker XRD equipment (model: D8 Discover with GADDS, Karlsruhe, Germany) using Ni-filtered CuKα radiation (λ = 0.154 nm). Raman spectra were collected by a Raman spectroscopy (model: Renishaw, London, UK) using a 532 nm laser. N_2_ and CO_2_ sorption were measured by a BET surface area analyzer (model: TriStar II 3020, Georgia, USA) operated at −196.4 °C and 0 °C, respectively, and the pore size distribution was calculated by density functional theory model.

### 2.5. Permeation Measurements

The gas separation performances of all membranes were conducted using a constant volume system with a 38.6 mL volume of membrane pool (Lanmo Tech, Changsha, China). All gas tests were conducted in the order of N_2_, CH_4_, O_2_, H_2,_ and CO_2_ at a temperature of ~20 °C and a feed pressure of ~3.8 bar. N_2_ was firstly ventilated into the system and experienced an original test to remove other impurity gases in the pipeline before the formal test. Moreover, to prevent aging, CMSMs were tested within one day after preparation. Permeability was calculated using Equation (1).
(1)$$p_{i}\;=\;\frac{273.15}{76}\frac{V}{AT}\frac{l}{\Delta p_{i}}\left(\frac{dp}{dt}\right)$$
where P_i_ refers to permeability of ‘i’ gas in Barrer (1 Barrer = 1 × 10^−10^ cm^3^ (STP) cm/cm^2^ s cmHg). V, A, T, l, and Δp_i_ are downstream volume (cm^3^), test area (cm^2^), temperature (K), the thickness of membranes, and pressure difference on both sides of membranes. The value of 273.15 and 76 are the temperature (K) and the operation pressure (cmHg) at the standard state, respectively. (dp/dt) is the slope of the pressure and time (s). The ideal selectivity (α_ij_) of the pure gas was calculated using Equation (2).
(2)$$\alpha_{ij}=\frac{p_{i}}{p_{j}}$$

## 3. Results

### 3.1. CA/PI Composite Membranes

Figure 1 shows the attenuated total reflectance–Fourier transform infrared (ATR-FTIR) spectra of 6FDA-DAM/DABA (3:2) polyimide (PI) membrane and CA/PI composite membranes. The typical characteristic imide peaks of neat PI at 1785 cm^−1^, 1725 cm^−1^, 720 cm^−1,^ and 1367 cm^−1^ are observed, which are corresponding to symmetric C = O stretching vibration, asymmetric C = O stretching vibration, deformation of imide ring band of OC-N-CO, and -C-N-C stretching, respectively. Moreover, the peaks at 1250 cm^−1^, 1203 cm^−1^, and 1150 cm^−1^ are assigned to the C-F bond of 6FDA moiety [34]. Moreover, in the pure CA, the vibration band in the range of 3750–3550 cm^−1^ corresponds to -OH stretching. Additionally, characteristic peaks of pure PI and CA are observed in all blend polymer membranes, proving that the physical blending is successful.

The SEM and optical images of CA/PI membranes are summarized in Figure 2. While the neat PI membrane is homogeneous, CA/PI composite membranes exhibit phase separation, a typical sea (PI)-island (CA) structure. Additionally, along with the increase of CA content, the volume fraction of CA domains increases. At a CA content of 0.5 or 1 wt.%, the size of CA domains is relatively small (less than 2 μm); when CA content is equal and higher than 5 wt.%, the size of CA domains significantly increases at a higher content, which is consistent with previous studies [26]. Although the CA/PI composite membranes exhibit distinct phase separation in the matrix, their surfaces are smooth and flawless.

Based on Figure 2, it is clear that phase separation occurs when CA and PI are mixed. However, it is not concluded that whether CA and PI are distinctly immiscible or partially miscible. The glass transition temperature (T_g_) and chain activation energy are good indicators for judging the compatibility of polymer blends. The glass transition of PI is measured by DMA, the frequency-dependent tan(δ) curves are plotted in Figure 3 and the T_g_ is summarized in Appendix A in Appendix A. For the glass transition of PI, the temperature becomes higher with the addition of CA. According to Kissinger’s equation, the corresponding activation energy (E_a_) could be calculated (Figure 3f). For the neat PI membrane, the E_a_ is 249.6 KJ/mol, with the addition of 0.5 wt.% and 1 wt.% CA, the E_a_ increases to 269.1 KJ/mol and 293.0 kJ/mol, respectively. The significant increase (up by 17.4%) of E_a_ suggests that the mobility of PI chains is strongly restricted by CA chains. At a CA content of 5 wt.% or higher, the E_a_ decreases to about 251.0 KJ/mol, which is close to the E_a_ of neat PI and proves that PI and CA are totally immiscible. These changes indicate that PI and CA chains could be partially miscible at a low CA content; the same phenomena were observed in other polymer composites in our previous research [35]. The chain interactions between PI and CA are expected to affect the gas diffusion in the PI matrix.

The gas permeation behaviors of CA/PI composite membranes are measured in the sequence of H_2_, CO_2_, O_2_, N_2_, and CH_4_. Figure 4a shows the gas permeabilities of different CA/PI membranes. While the PI membrane shows the highest gas permeability, the CA membrane exhibits the lowest gas permeability. The BET surface area and total pore volume of PI are 178 m^2^/g and 0.15 cm^3^/g (Appendix A in Appendix A), respectively. By comparison, the BET surface area and total pore volume of CA are too low to be measured. It is expected that CA domains would block the gas diffusion, lead to a longer diffusion path, and result in lower gas permeability. Figure 5 shows the scheme of gas diffusion in CA/PI membranes. At a CA content of 5 wt.% and higher, the CA islands would block the gas diffusion and lead to a longer path. Along with the increase of CA content from 0 to 5 wt.%, to 10 wt.%, and to 20 wt.%, the hydrogen permeability decreases from 130 Barrer to 94 Barrer, to 84 Barrer, and to 78 Barrer, respectively. The same trend is observed for the permeabilities of other gases. At a higher CA content, the diffusion path in the PI matrix is expected to be longer, which leads to a lower apparent gas flux, while for CA/PI membranes with CA contents of 0.5 and 1 wt.%, the gas permeability is much lower than other samples. The gas permeability of CA0.5 and CA1 membranes is only about 57% and 37% of the PI membrane. The DMA results suggest partial miscibility of CA in the PI matrix at a low content, and PI chain mobility is suppressed. Nitrogen sorption measurement (Appendix A in Appendix A) of CA1 membrane shows a much lower surface area (69 m^2^/g) and total pore volume (0.06 cm^3^/g), as compared with PI membrane, 178 m^2^/g, and 0.15 cm^3^/g, respectively. The same result is shown in Appendix A. The chain interactions between PI and CA clearly reduce the micropores and lead to a higher gas diffusion resistance, and the corresponding scheme is shown in Figure 5. As shown in Figure 5, the reduced gas permeability in CA/PI membranes is ascribed to different mechanisms depending on phase structures. At a CA content of 1 wt.% and lower, the partial compatibility of the CA chain in the PI matrix forms a denser and less porous structure and leads to a higher gas diffusion resistance. That is why the gas permeability of CA0.5 and CA1 is much lower than PI. By comparison, at a CA content of 5 wt.% and higher, the complete phase separation leads to a sea (PI matrix) and island (CA domains) structure. Since the CA domain has a very low gas permeability, it acts as a blocker. Along with the increase of CA content, the gas diffusion path in the PI matrix becomes narrower and longer, which results in a higher diffusion resistance. In Figure 4a, it could be seen that the gas permeability decreases in the sequence of PI, CA5, CA10, and CA20. Additionally, for PI and CA/PI membranes, the gas selectivity (Figure 4b) of H_2_/CO_2_, H_2_/N_2_, and H_2_/CH_4_ barely changes, which indicates that the interactions between PI and CA do not affect the molecule sieving behaviors of PI.

### 3.2. Heat-Treated CA/PI Membranes

CA is a thermally degradable polymer and has a relatively low char yield. The TGA curves of CA and PI in nitrogen are shown in Figure 6a. The major weight loss of CA occurs in between 350 to 380 °C, and the residual weight is lower than 20 wt.%. In contrast, the decomposition of PI begins at about 430 °C, and the weight loss is about 6 wt.% even after heat treatment up to 500 °C [9].

The derivative curves of thermal weight loss are plotted in Figure 6b. The maximum weight loss temperatures of CA and PI are about 370 °C and 550 °C, respectively. Since the decomposition temperature of CA is much lower than PI, it acts as a sacrificial agent and could be used for creating pores in the PI matrix [36,37]. With the addition of 0.5 and 1 wt.% CA, the maximum weight loss temperature of PI decreases from 550 °C to 546 °C, and to 543 °C, respectively. At a CA content of 5 wt.% and higher, the maximum weight loss temperature of PI is close to the pure PI. Above peak temperature, changes indicate that the residual char of CA affects the decomposition of PI for CA0.5 and CA1 membranes but has no direct effect at a CA content of 5 wt.% or higher, which is consistent with the phase behaviors of CA/PI membranes detected by DMA. The SEM images of heat-treated membranes are shown in Figure 7, and the corresponding optical images are summarized in Appendix A in Appendix A. After heat treatment at 350 °C, the neat PI membrane has a solid structure. Along with the addition of more CA, both the pore size and total volume percentage increase. The formation of pores is consistent with the phase diagram of CA/PI composite membranes, shown in Figure 2.

Based on the TGA data in Appendix A, about 17 wt.% residual char is formed for CA after heat treatment. An elemental scan was carried out to examine the chemical composition of the CA char. The O/C ratios of heat-treated membranes are plotted in Figure 8a. It is clear that the O/C ratio increases when CA content increases. The F/C ratios of the inner surface of pores and PI matrix of heat-treated CA20 are shown in Figure 8b. Since PI contains Florine elements, the F/C ratio of the PI matrix is about 0.08. By comparison, the F/C ratio of the inner surface of the pores is lower than 0.02, which demonstrates that the decomposed CA leaves a thin layer of residual chars on the inner surface of the pores.

The gas permeability measurements results are summarized in Figure 9. The gas permeability of heat-treated PI is about 30% higher than the neat PI, possibly due to the crosslinking of the PI chain [38]. The porosity of PI-350 and CA1-350 were measured by CO_2_ sorption (Appendix A in Appendix A). The PI-350 exhibits a larger sorption capacity and the pore widths are distributed in 0.6~0.65 and 1~1.05 nm, with an incremental micropore volume higher than 0.036 cm^3^/g. Same as the change of the gas permeability of CA/PI membranes, the addition of 0.5 and 1 wt.% CA in PI leads to a much lower flux after heat treatments. The porosity measurements (CO_2_ sorption) of CA1-350 show a pore size distribution mainly in the range of 0.4~0.85 nm and an incremental micropore volume less than 0.007 cm^3^/g (Appendix A in Appendix A), which is obviously lower than PI-350. These results prove that the addition of a small amount of CA causes a significant reduction of fine pores and leads to much lower gas permeability. In contrast, for CA content that is 5 wt.% or higher, the gas permeability increases at a higher CA content. The XRD curves of CA before and after heat treatment are plotted in Figure 9d. The crystallization peaks of CA at around 13°, 17°, and 22.6° are ascribed to (−110), (10-1), and (002) planes [39,40]. Along with the decomposition of CA, the semi-crystalline structure is transformed into amorphous chars. The N_2_ adsorption measurement of CA powder exhibits a type-I sorption curve, a BET surface area of 386 m^2^/g, and an incremental micropore volume of 0.04 cm^3^/g (Appendix A in Appendix A and Figure 9c). The formation of porous structures of CA after heat treatment contributes to the improvement of gas permeability after heat treatment. However, it must be noted that even though CA/PI membranes contain many macro-pores after heat treatments, and these pores are expected to lead to a higher gas permeability, the gas permeability of CA20-350 is still lower than PI-350. According to the above data, a thin char layer (decomposed CA) is formed in the inner surface of the pores, and the layer must reduce the gas diffusion, as compared with PI-350, and result in lower gas permeability. There are comprehensive effects for the gas permeation of heat-treated CA/PI composite membranes, including pore formation and resistant thin char layer. Additionally, the gas selectivity remains the same regardless of the addition of CA (Figure 9b).

A scheme for the gas diffusion in heat-treated PI and CA/PI membranes is summarized in Figure 10. For CA/PI composite membrane with a CA content of 1 wt.% and lower, the PI and CA chains are partially compatible and cause the formation of a denser structure than neat PI after heat treatment at 350 °C. In this case, the entire matrix exhibits high resistance for gas diffusion and leads to significantly lower gas permeability. For CA/PI composite membrane with a CA content of 5 wt.% and higher, the composite shows complete phase separation, and the properties of the PI matrix are not affected by CA before and after heat treatment. However, the CA does not completely degrade and leaves a thin layer of residual char on the inner surface of voids. Even though the formation voids reduce the gas diffusion path length, the thin char layer must have a higher resistance than the heat-treated PI matrix and block the gas diffusion. Thus, after heat treatment at 350 °C, the overall gas permeability of CA/PI membranes becomes higher at a higher CA content (5 wt.% and higher) but is still lower than PI.

### 3.3. CA/PI-Based Carbon Molecule Sieve Membranes (CMSMs)

The PI and CA/PI membranes were further pre-carbonized at 600 °C. The surface and cross-sectional images of the obtained CMSMs are summarized in Figure 11. The formation of pores shows similarity to CA/PI membranes before and after heat treatment at 350 °C and suggests that the phase structures of the composite membranes are inherited from the precursor membranes. The cross-sectional images of the fractured surface of CMSMs become smoother than the precursor membranes since the PI matrix becomes more brittle after carbonization.

The gas permeability of various CMSMs is shown in Figure 12a. For PI, the gas permeability significantly increases after thermal treatments and becomes much higher after carbonization at 600 °C than heat treatment at 350 °C. Additionally, unlike precursor and heat-treated membranes, the CA/PI-based CMSMs show higher permeability than PI-based CMSM (Figure 12a). While the H_2_ and CO_2_ permeability values of PI-600 are 3420 and 1659 Barrer, respectively, they are close to CA0.5-600 and CA1-600, and they significantly increase to 5309 and 3200 Barrer (>54% improvement), respectively, for CA10-600. Additionally, the H_2_/N_2_, H_2_/CH_4_, and H_2_/CO_2_ selective ratios are about 42, 42, and 1.6, respectively, for all CMSMs, which breaks through the Robeson’s upper bound presented in 2008 (Appendix A in Appendix A). After carbonization at 600 °C, the BET surface area and total pore volume of PI increase from 178 m^2^/g and 0.15 cm^3^/g to 743 m^2^/g and 0.40 cm^3^/g, respectively (Appendix A in Appendix A), which is higher than the BET surface area (407 m^2^/g) and total pore volume (0.29 cm^3^/g) of CA-600. The consistent result is reflected on N_2_ sorption isotherms curves (Appendix A in Appendix A). The higher gas permeability of these CMSMs could be ascribed to higher surface area and pore volume. Based on WAXD, the peak at 45° corresponding to graphite (001) crystal plane peak becomes more obvious after carbonization at 600 °C (Appendix A in Appendix A) [41,42], which suggests that the in-plane cross-linking leads to larger graphitic structures. Additionally, the Raman I_D_/I_G_ ratios of PI-600, CA1-600, and CA10-600 membranes are 0.87, 0.87, and 0.88, respectively (Appendix A in Appendix A), indicating no obvious difference after carbonization. However, the pore size distribution and Raman spectra show that the PI-600 has a narrower pore distribution and higher graphitization degree than CA-600 (Figure 12c,d), which suggests that the formed carbon structure of PI is more compact and the micropore is relatively smaller after carbonization at 600 °C [43,44], and the pyrolytic product of CA is not provided enhanced selectivity with a more open structure. Additionally, the H_2_/N_2_ perm-selectivity of PI-600 is about 53, much higher than PI-350 or PI (~32), which is mostly attributed to the pyrolytic PI matrix.

Therefore, the improved gas permeability is caused by both the formation of macro-pores and the structural evolution of the PI matrix after pyrolysis. The macro-pores act as the fast transport channels and reduce the gas transport resistance (Figure 13). The gas permeability of CA20-600 is lower than that of CA10-600, but its selectivity is slightly higher, which might be caused by more oxygen content introduced by CA decomposition [5,45]. Ultimately, the CA/PI-based CMSMs exhibit unique properties different from other pore-forming agents due to the higher carbon yield, including PVP and PEG, and confirm that the phase structures of polymer blend play a decisive role in the permeability of polymer and carbon membranes. For polymer membrane, the partial compatibility of PI and CA leads to a high gas diffusion resistance, and in addition, the CA domains block gas diffusion. These two factors cause the gas permeability of CA/PI composite membranes even lower than PI membranes. After heat treatment at 350 °C, the partially compatible CA chains in the PI matrix result in lower micro-porosity and also reduce the gas permeability, as compared with the PI membrane. Only after carbonization at 600 °C, the formed carbon structures of CA are similar to PI in the view of micro-porosity and BET surface area, the formation of macro-pores comes to play roles in improving the gas permeability and leads to over 50% improvements. For gas selectivity, the characters of the PI matrix are decisive.

## 4. Conclusions

In summary, both the chemical and physical structures of a polymer-based membrane are essential for its gas separation performances. However, thus far, most studies focus on how the chemical structures of a polymer affect its performance. In this study, a sacrificial component, CA, is added into PI-based membranes. With the addition of CA, the gas flux of PI membrane significantly decreases, especially for low CA content samples, less than 1 wt.%. By making structural analysis, it is found that a small amount of CA is miscible in the PI matrix when CA content is ≤1 wt.%, and traction between CA and PI chains leads to a higher PI chain activation energy and lower porosity. Additionally, the CA phase has very low gas permeability and leads to a longer gas permeation path in phase-separated membranes. Both above factors reduce the gas permeability of the CA/PI composite membrane. After heat treatment at 350 °C, most CA decomposes, and the phase-separated CA domains form pores. When CA content is ≤1 wt.%, the miscible CA chain in the PI matrix leads to less porous structures, as compared with heat-treated PI membranes, and leads to lower gas permeability. In contrast, when CA content is ≥5 wt.%, the residual char from the decomposition of CA coats on the inner surface of pores blocks the gas diffusion and leads to a gas permeability even lower than heat-treated PI membrane. Normally, the formation of pores is expected to reduce the gas diffusion path and improve the gas flux. However, there is always some residual chars when sacrificial component decomposes, and these chars might play various roles, either blocking or low resistance. After further carbonization at 600 °C, the CA char is further carbonized, and the structural differences between carbonized PI and CA become small. The formation of pores takes effect, and the gas permeability significantly increases when more and more pores are formed in CMSMs. The CA10-600 shows the highest properties with a H_2_ permeability of 5300 Barrer and a H_2_/N_2_, H_2_/CH_4_, and H_2_/CO_2_ permselectivity of 42, 42, and 1.6, respectively. By comparison, the control sample, PI-600, only has a H_2_ permeability of 3300 Barrer.

## Figures and Tables

**Figure 1 membranes-11-00618-f001:**
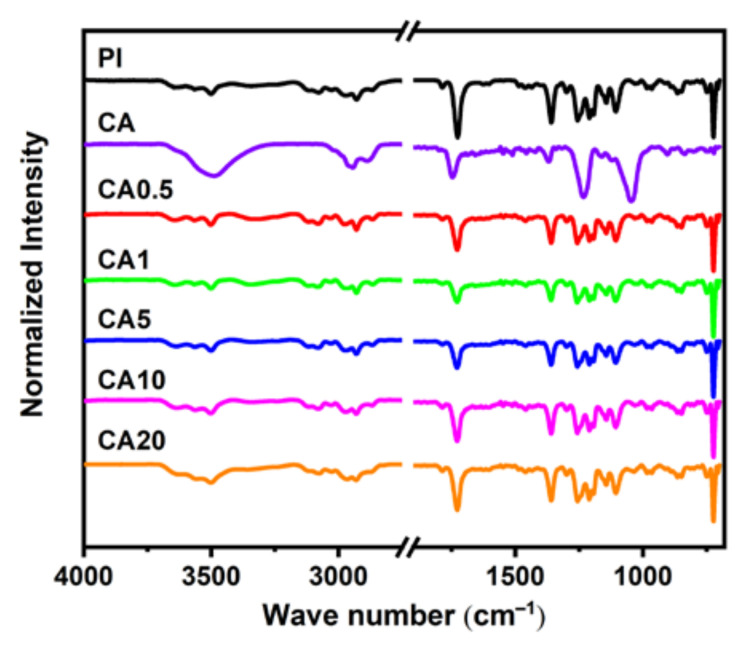
FT-IR spectra of pure CA, pure PI, and composite membranes.

**Figure 2 membranes-11-00618-f002:**
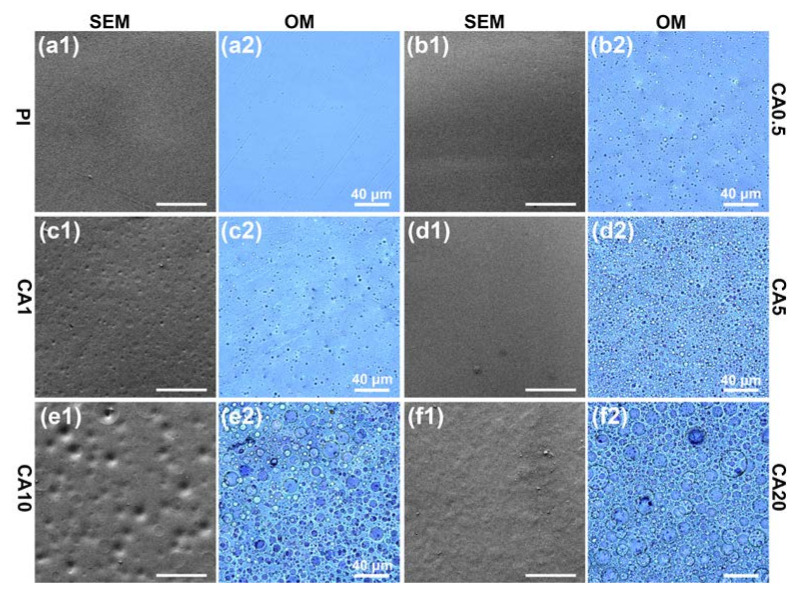
SEM and optical images of composite membranes: (**a**1–**f**1) and (**a**2–**f**2) are polymer membranes of PI, CA0.5, CA1, CA5, CA10, CA20, respectively.

**Figure 3 membranes-11-00618-f003:**
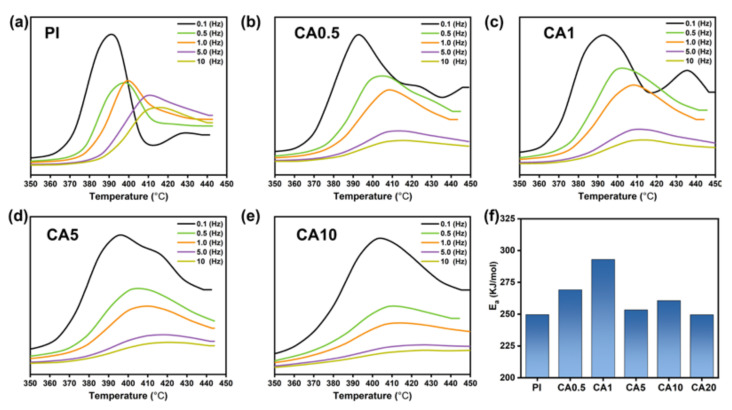
(**a**–**e**) tan(δ) curves of CA/PI membranes monitored by DMA; (**f**) activation energies.

**Figure 4 membranes-11-00618-f004:**
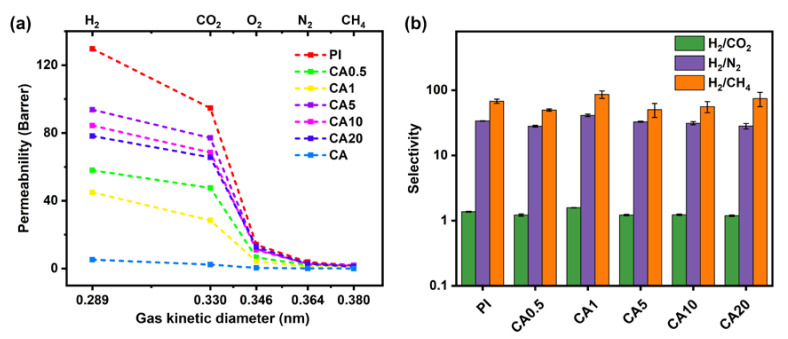
Gas permeability (**a**) and selectivity (**b**) of composite membranes with different content of CA.

**Figure 5 membranes-11-00618-f005:**
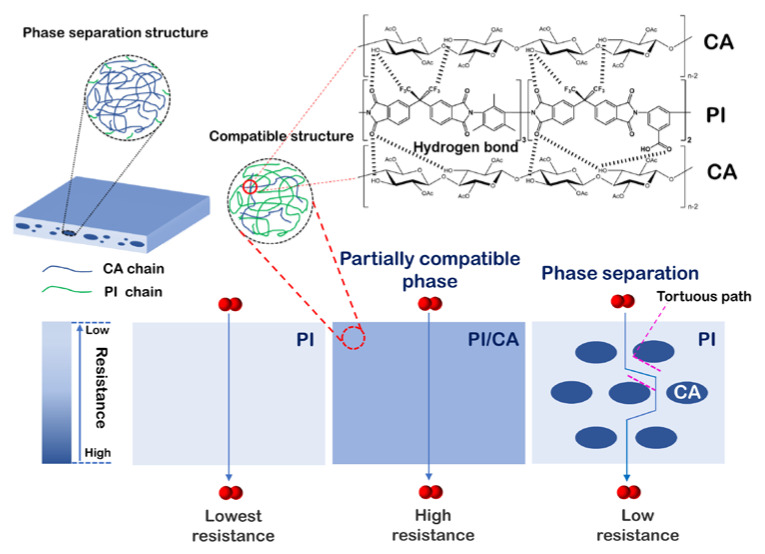
The mechanism of gas permeability of different phase structures for composite membranes.

**Figure 6 membranes-11-00618-f006:**
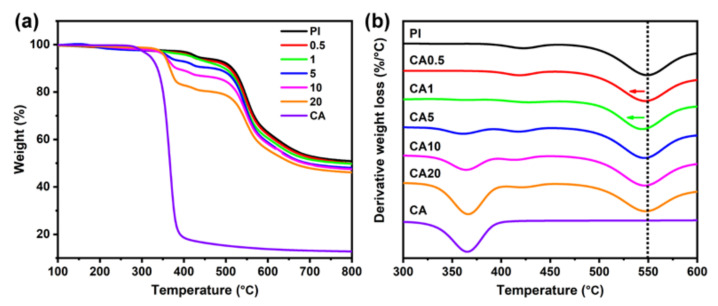
Thermal gravimetric analysis (TGA) of composite membranes: (**a**) weight loss curves; (**b**) derivative weight loss curves. The test is carried out at a 10 ^○^C/min heating rate in N_2_.

**Figure 7 membranes-11-00618-f007:**
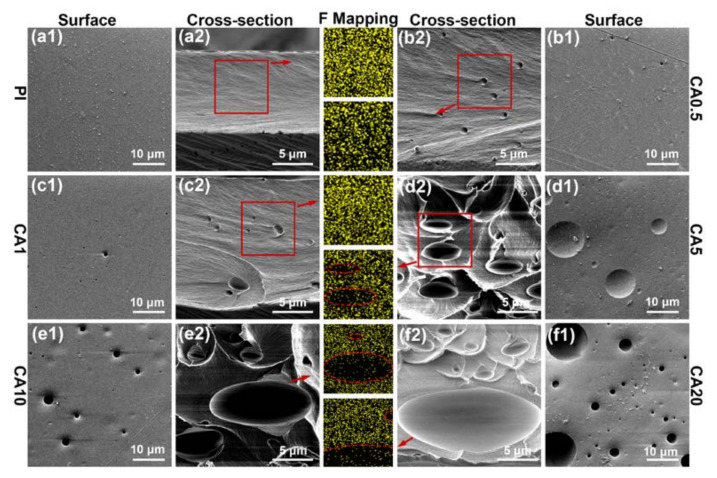
SEM images of heat-treating membranes: (**a**1–**f**1) and (**a**2–**f**2) are surface and cross section of PI, CA0.5, CA1, CA5, CA10, CA20 membranes, respectively. The middle images are selected area EDXS maps (F single: yellow).

**Figure 8 membranes-11-00618-f008:**
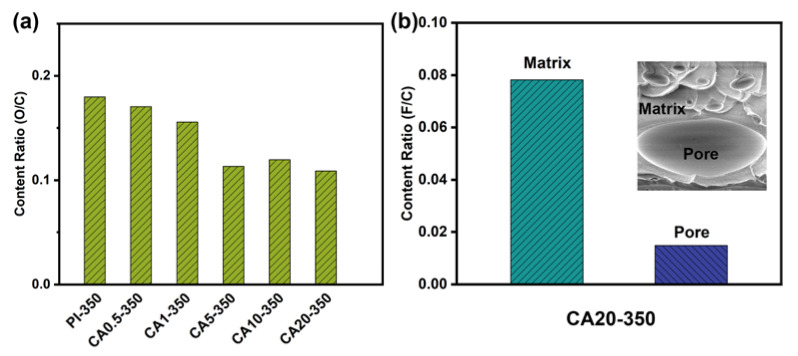
Element analysis of composite membranes at 350 °C: (**a**) content ratio (O/C) of matrix; (**b**) content ratio (F/C) of matrix and pores for CA20-350.

**Figure 9 membranes-11-00618-f009:**
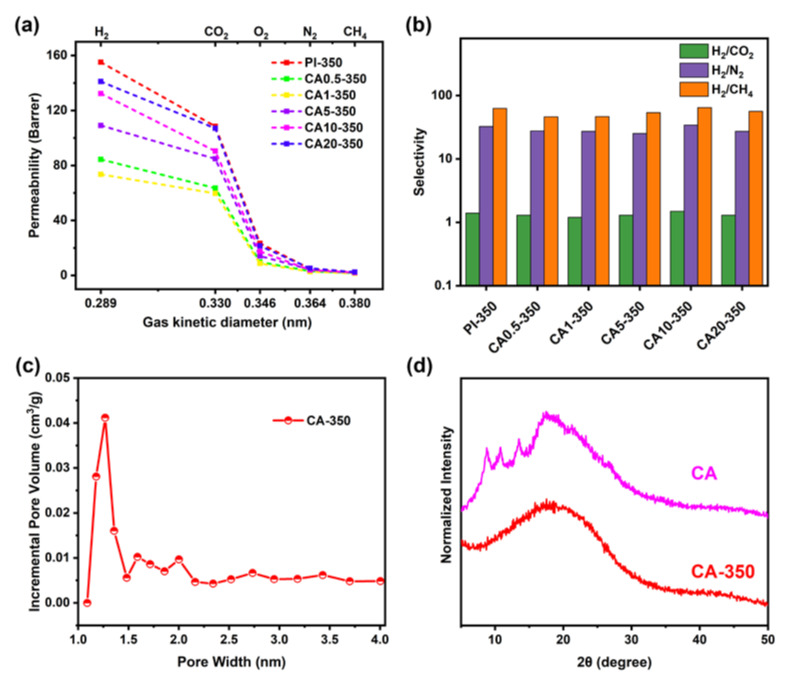
Gas permeability (**a**) and selectivity (**b**) of composite membranes after heat treatment at 350 °C; (**c**) pore size distribution curve based on N_2_ sorption and (**d**) WAXD patterns.

**Figure 10 membranes-11-00618-f010:**
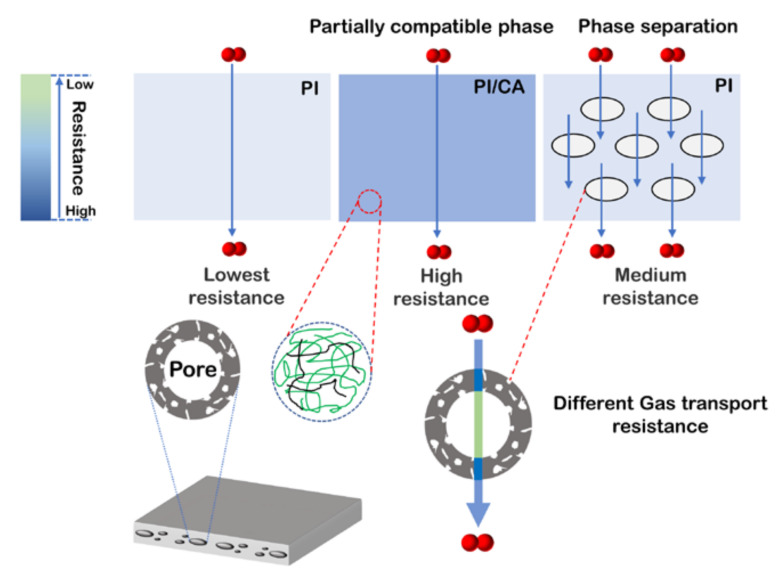
The mechanism of gas permeability of different phase structures for composite membranes after 350 °C treatment.

**Figure 11 membranes-11-00618-f011:**
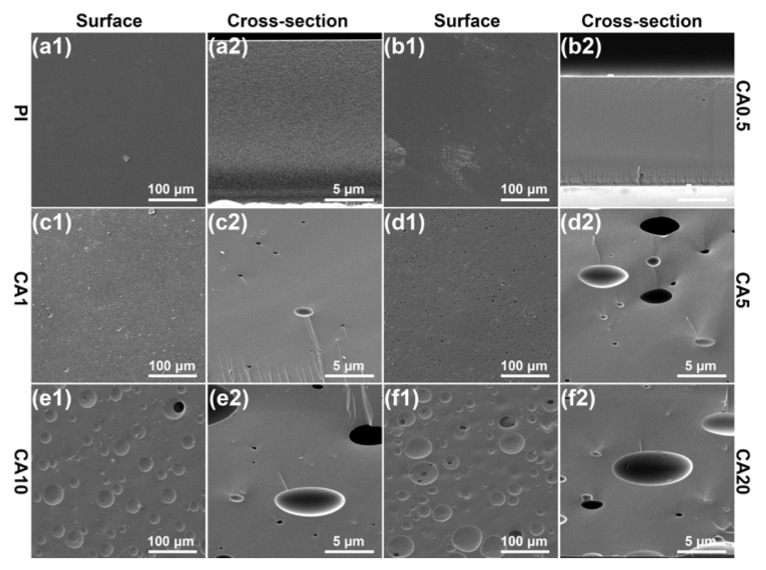
SEM images of CMSMs: (**a**1–**f**1) and (**a**2–**f**2) are the surfaces and cross sections of PI, CA0.5, CA1, CA5, CA10, CA20 CMSMs, respectively.

**Figure 12 membranes-11-00618-f012:**
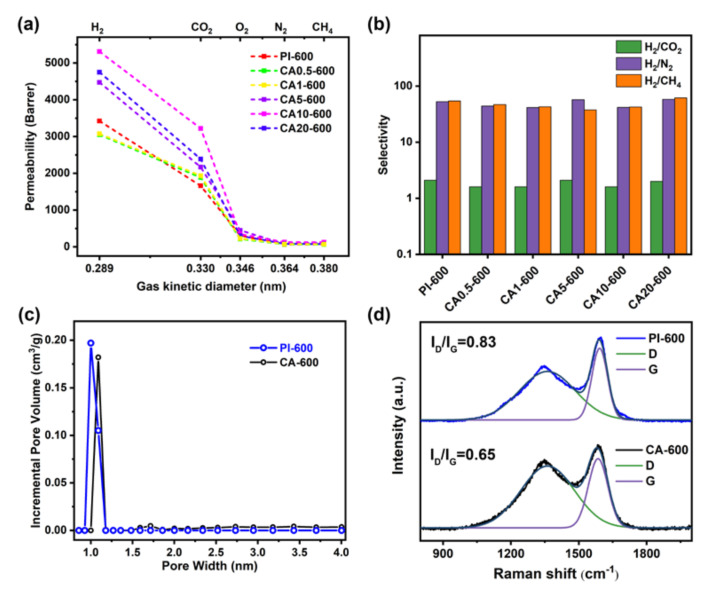
Gas permeability (**a**) and selectivity (**b**) of CMSMs; (**c**) pore size distribution curves based on N_2_ sorption and (**d**) Raman spectra.

**Figure 13 membranes-11-00618-f013:**
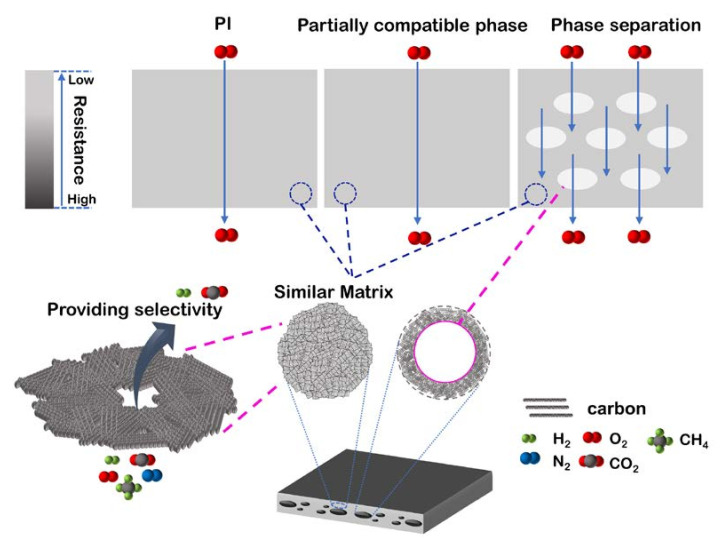
The mechanism of gas permeability of different phase structures for CMSMs.

**Table 1 membranes-11-00618-t001:** List of sample compositions and abbreviations.

Composite Membranes	350 °C Treatment	600 °C Treatment	Mass Fraction (wt.%)	
			PI	CA
PI	PI-350	PI-600	100	0
CA0.5	CA0.5-350	CA0.5-600	99.5	0.5
CA1	CA1-350	CA1-600	99	1
CA5	CA5-350	CA5-600	95	5
CA10	CA10-350	CA10-600	90	10
CA20	CA20-350	CA20-600	80	20
CA	CA-350	CA-600	0	100

## Data Availability

Not applicable.

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
