# Peer review of "The Phase Structural Evolution and Gas Separation Performances of Cellulose Acetate/Polyimide Composite Membrane from Polymer to Carbon Stage"

_membranes, 2021, doi:10.3390/membranes11080618_

Round 1

Reviewer 1 Report

In this work, the authors introduced cellulose acetate (CA) into polyimide (PI) as sacrificial phase to adjust the structure and gas separation performance from polymer to carbon. After heat-treatment at 650 degrees the CA10-600 showed highest separation properties with a H2 permeability of 5300 Barrer and a H2/N2, 393 H2/CH4 and H2/CO2 perm-selectivity of 42, 42 and 1.6, respectively.

This work is well written and all the figures are nicely presented. However, the article lacks of XPS study to shade some light on the interactions site and selectivity of gas transformation with in the framework. It is necessary to explore possible factors that are responsible for the interaction with the gases followed by separation.

With this additional analysis and discussion, this paper can be published.

Reviewer 2 Report

The authors presented the effect of blending and the heat treatment on the permeability and selectivity of the polymers. The authors carefully prepared the polymers and characterized by SEM, IR, TGA, Raman, gas sorption, and DMA instruments. In addition, the gas permeability and selectivity were properly analyzed for the blended polymers and the heat-treated polymers. The authors demonstrated the effect of CA blending in PI polymer by analyzing the blended polymers and their structure. In addition, the gas permeability of the carbonized polymers was also analyzed. Although permeability is not the best value, the analysis of the structure-property relation is one of the most important issues in gas permeable membrane development. Therefore, I recommend acceptance of the manuscript after minor revision.

  1. The authors may want to provide more details about the analytical condition of the materials. (IR, gas sorption, pore size distribution calculation, Raman and etc) Although the presented analysis is routine work, if there are no experimental details, it is difficult to reproduce the presented results.
  2. Although the authors mentioned that pore size distribution and BET surface area of PI-350 and CA1-350 were calculated using CO2 sorption data. However, no CO2 sorption data were presented in the manuscript or supporting information. The authors must carefully check and present the data.

Round 2

Reviewer 1 Report

The authors answered my comments without even tying the XPS analysis. However, I understand that there might be some issue in XPS analysis or facilities to access this during the COVID times. Considering this, I am willing to accept this modified version.